# Multisite Metagenomic Next-Generation Sequencing Improved Diagnostic Performance for Sepsis-Associated Lymphopenia Patients

Dongkai Li,[a] Wei Gai,[b] Jiahui Zhang,[a] Wei Cheng,[a] ⓘ Na Cui,[a] Hao Wang[c]

[a]Department of Critical Care Medicine, Peking Union Medical College Hospital, Beijing, China
[b]WillingMed Technology (Beijing) Co., Ltd., Beijing Economic-Technological Development Area, Beijing, China
[c]Department of Critical Care Medicine, Beijing Jishuitan Hospital, Beijing, China

Dongkai Li and Wei Gai contributed equally to this article. Author order was determined by type of contribution.

**ABSTRACT** A precise and efficient microbiological diagnosis is essential for sepsis. Metagenomic next-generation sequencing (mNGS) is a novel technique for the diagnosis of infectious diseases, but its current application in multisite sampling and interpretation remains controversial. Therefore, this study was undertaken to evaluate the reliability of multisite mNGS tests and the efficiency of plasma mNGS based on lymphocyte subset counts. A prospective observational study was performed on the intubated patients with sepsis-associated lymphopenia from January 2020 to February 2022. During the study period, data on 71 patients with sepsis-induced lymphopenia were collected. Among the 125 mNGS tests, 95 were positive for pathogens, whereas of the 166 conventional microbiological tests (CMTs), 91 were positive. The comparison showed that 38 patients (53.5%) had at least one matched pair of plasma mNGS and CMT results, while for multisite sampling, 47 patients (66.2%) had at least one. Lymphocyte subset analysis showed that T lymphocyte (577 $\pm$ 317 versus 395 $\pm$ 207, $P = 0.005$) and CD4$^+$ T lymphocyte (333 $\pm$ 199 versus 230 $\pm$ 120, $P = 0.009$) counts were lower in the matched group. According to receiver operating characteristic (ROC) analysis, a CD4$^+$ T lymphocyte count lower than 266 cells/mm³ was predictive of a match result. For sepsis-associated lymphopenia patients, we found that multisite mNGS tests showed a higher positivity rate. With plasma mNGS, a lower CD4$^+$ T lymphocyte count predicted a better match result with CMT. The lymphocyte subset analysis may promote the clinical interpretation of mNGS results.

**IMPORTANCE** This study was undertaken to evaluate the reliability of pathogenic diagnoses based on multisite mNGS detection at the clinically suspected sites and to analyze the efficiency of plasma mNGS detection based on lymphocyte subset counts in patients with sepsis-associated lymphopenia.

**KEYWORDS** sepsis, lymphopenia, metagenomic next-generation sequencing, mNGS

S epsis represents a global health care problem in intensive care units (ICUs) and is the most common cause of in-hospital and ICU mortality (1). Although the onset and progression of sepsis are substantially heterogeneous, the occurrence of severe immunosuppression, with which sepsis-associated lymphopenia is commonly associated, is frequently observed in septic patents and was demonstrated to be significantly correlated with deteriorative clinical outcomes (2, 3). The early identification of pathogens is crucial for implementing effective pathogen-targeted therapies to improve outcomes (4). However, pathogenic diagnosis in sepsis-associated lymphopenia patients is challenging because they are susceptible to common pathogens as well as numerous opportunistic pathogens. The low detection positivity rate

Address correspondence to Na Cui, pumchcn@163.com, or Hao Wang, newwanghao@hotmail.com.

The authors declare no conflict of interest.

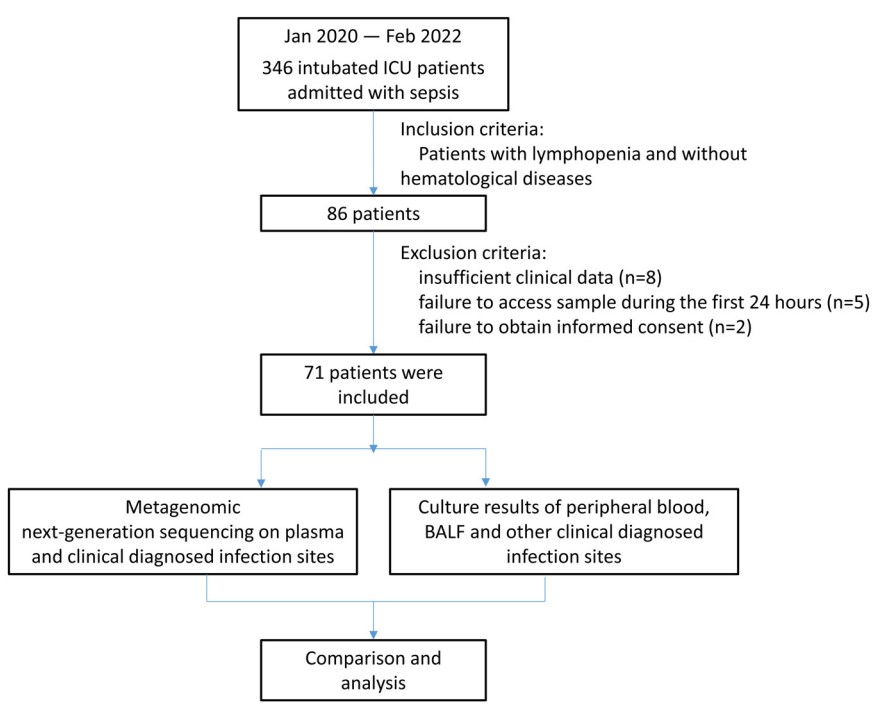

FIG 1 Flowchart of the prospective study.

and time-consuming processing of conventional microbiological tests (CMTs) are key problems faced by clinicians.

Metagenomic next-generation sequencing (mNGS) has been widely used in clinical settings in recent years. Compared with CMT methods, mNGS offers the significant advantages of short turnaround times and unbiased analyses (5). The cost of mNGS has fallen greatly since its advent in 2004, and mNGS has become a fundamental platform for microorganism detection across multiple categories of body fluid (6). Because of the unique characteristics and overall severity of infections in sepsis-associated lymphopenia patients, rapid and accurate diagnostic methods are needed. In this study, we attempted to evaluate the reliability of pathogenic diagnoses based on multisite mNGS detection at the clinically suspected sites and to analyze the efficiency of plasma mNGS detection based on lymphocyte subset counts in patients with sepsis-associated lymphopenia.

## RESULTS

**Patient characteristics.** During the study period, 346 intubated patients were admitted to the ICU with sepsis diagnosis, and 86 of the patients had sepsis accompanied with new-onset sepsis-induced lymphopenia. Five patients were excluded for insufficient clinical data, three for not surviving for at least 24 h, five for not providing a sample during the first 24 h, and two for failure to provide informed consent. Finally, 71 patients were included in the study, as shown in Fig. 1. The distribution of sampling sources for mNGS and CMT is shown in Fig. 2, and the distribution of the clinically suspected infection sites is shown in Fig. S1 in the supplemental material. As to the types of infection diagnosed, 17 had a pulmonary infection and 10 had an intra-abdominal infection, while 34 patients presented with at least two clinically suspected infection sites. The clinical characteristics, and whether the results of mNGS detection matched those of the CMT method in the corresponding infection sites, are shown in Table S1.

**Distribution of detected pathogenic microorganisms.** Overall, among the 125 samples that underwent mNGS, 95 were positive for pathogens, whereas of the166 samples that underwent the CMT method, 91 were pathogen positive. Using standard microbiological and clinical practices, the pathogenic organisms detected by mNGS were classified as shown in Fig. 3. For both plasma and bronchoalveolar lavage fluid (BALF) sources, mNGS detected more Gram-negative than Gram-positive pathogens. The most frequently detected organisms

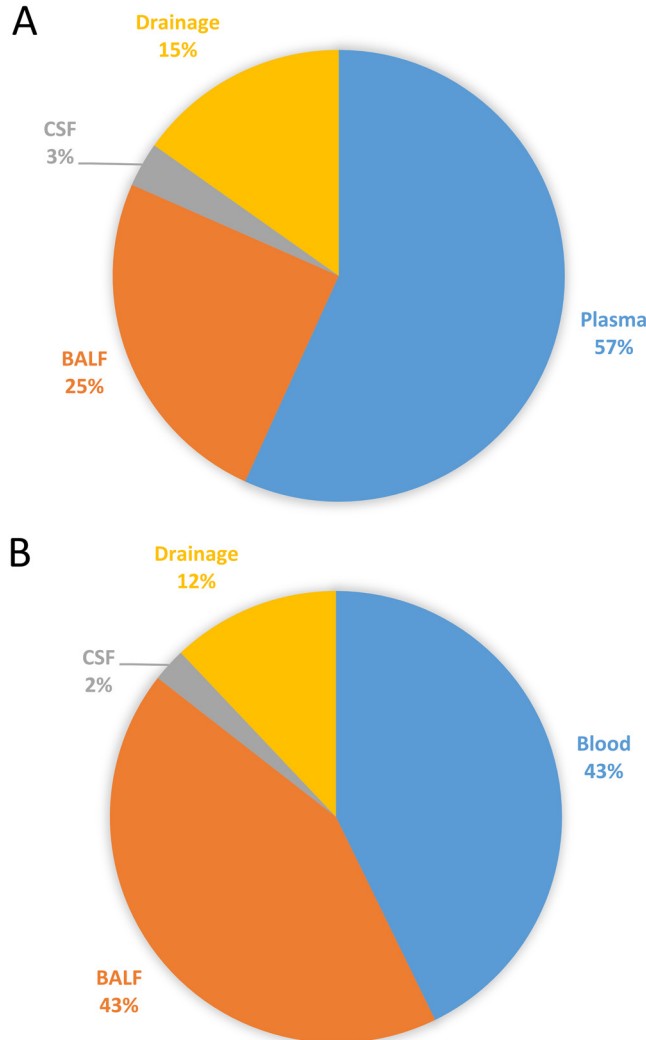

**FIG 2** Sampling sources analyzed by mNGS and CMT and distribution of clinically suspected infection sites. (A) Distribution of mNGS sampling sources. (B) Distribution of CMT sampling sources.

included the Gram-negative *Klebsiella*, *Pseudomonas aeruginosa*, and *Escherichia coli* and the Gram-positive *Staphylococcus*, *Enterococcus*, and *Streptococcus*. The mNGS detection results also included 54 strains of anaerobic bacteria, 23 of which were collected from plasma and 25 from drainage fluid, while the CMT method was unable to identify any anaerobic bacteria. For all four categories of clinically suspected infection sites (plasma/blood, BALF, cerebrospinal fluid [CSF], and drainage), the positive rates were higher using mNGS than CMT, as shown in Fig. 4A. A comparison of reporting times between the mNGS and CMT methods showed that the former identified the pathogens in a shorter time (26.6 $\pm$ 3.2 h for mNGS versus 80.5 $\pm$ 28.3 h for blood culture; 50.1 $\pm$ 18.2 h for CSF; 71.2 $\pm$ 23.8 h for drainage, $P < 0.05$; and 34.2 $\pm$ 13.5 h for BALF, $P = 0.324$), as shown in Fig. 4B.

**Comparison of metagenomic next-generation sequencing and conventional microbiological tests.** To evaluate the consistency of results between the mNGS and CMT methods, we compared the detection results of plasma mNGS, infection site mNGS, and infection site CMT methods. In the enrolled cohort, 45 cases (63.4%) were reported to be positive in both plasma mNGS and CMT methods at the clinically suspected infection sites, while 50 cases (70.4%) were reported to be positive with both the on-site mNGS and CMT methods, as shown in Fig. 5A and B. The comparison also showed that 38 patients (53.5%) had at least one matched pair of plasma mNGS and CMT results, including 17 cases (23.9%) with a partial match (the results for at least one pathogen matched) and 19 cases

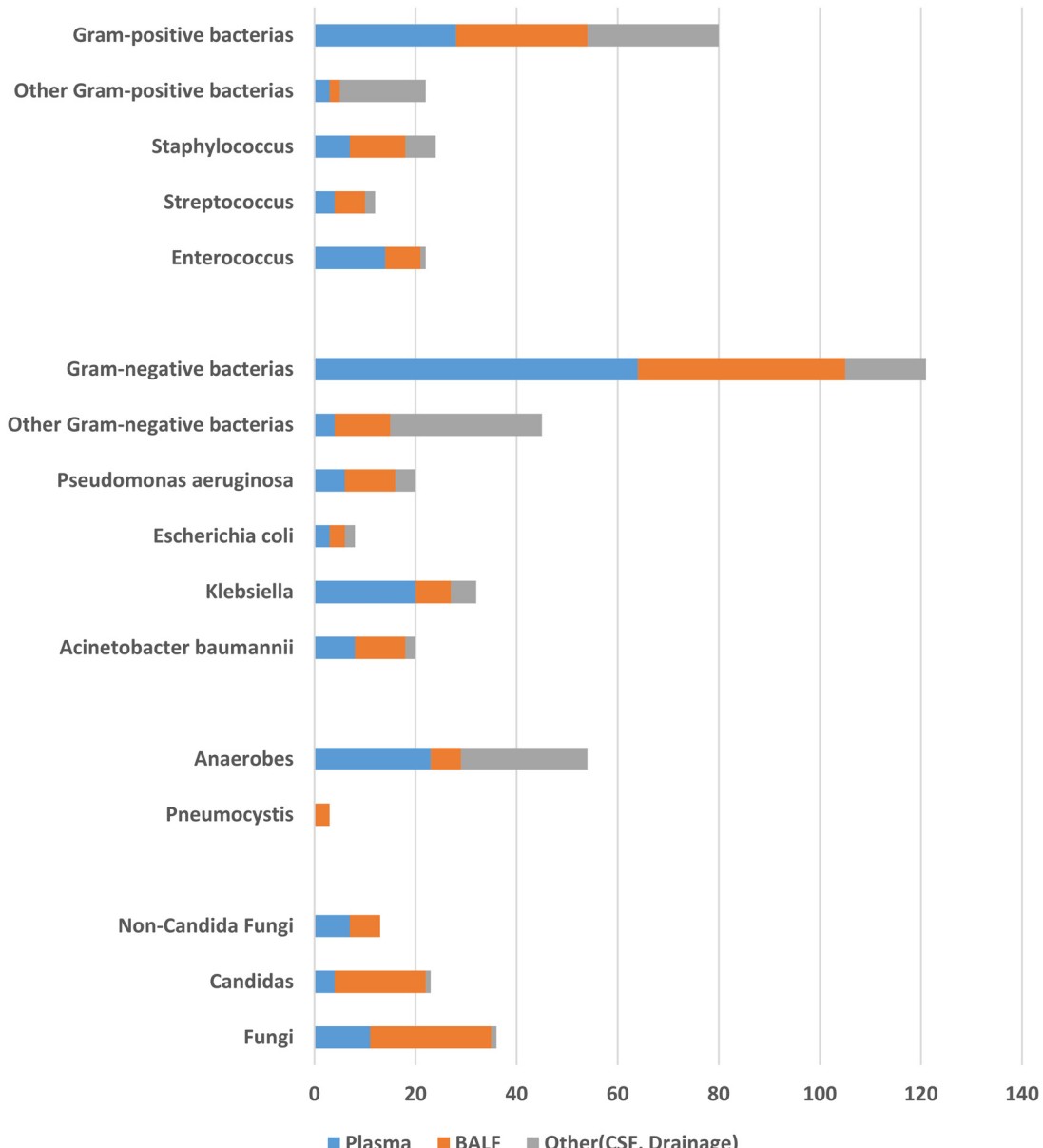

**FIG 3** Distribution of mNGS pathogens identified across different sampling sources. BALF, bronchoalveolar lavage fluid. CSF, cerebrospinal fluid.

(26.8%) of a complete match. While for the clinically suspected infection sites, the mNGS and corresponding CMT test results were better matched, there were 23 cases (32.4%) of a partial match and 24 cases (33.8%) of a complete match, as shown in Fig. 5C. With CMT method as a comparative method, the positive percent agreement and negative percent agreement between blood culture and plasma mNGS were 81.6% (95% CI: 66.6 to 90.8%) and 54.5% (95% CI: 38.0 to 70.2%), as shown in Tables S2 and S3.

**Lymphocyte subpopulation and metagenomic next-generation sequencing.** To further ascertain any potential factors associated with the matched results between plasma mNGS and CMT for the clinically suspected infection sites, we compared the differences in immune parameters, including inflammatory mediators and lymphocyte subsets, between the two groups, as shown in Table S1. No significant differences were identified between the two groups in age, sex, APACHE II score, SOFA score, or white blood cell count at ICU admission ($P > 0.05$), while the unmatched group showed significantly higher lymphocyte counts than the matched or partially matched group (761 $\pm$ 359 versus 548 $\pm$ 247 cells/mm$^3$,

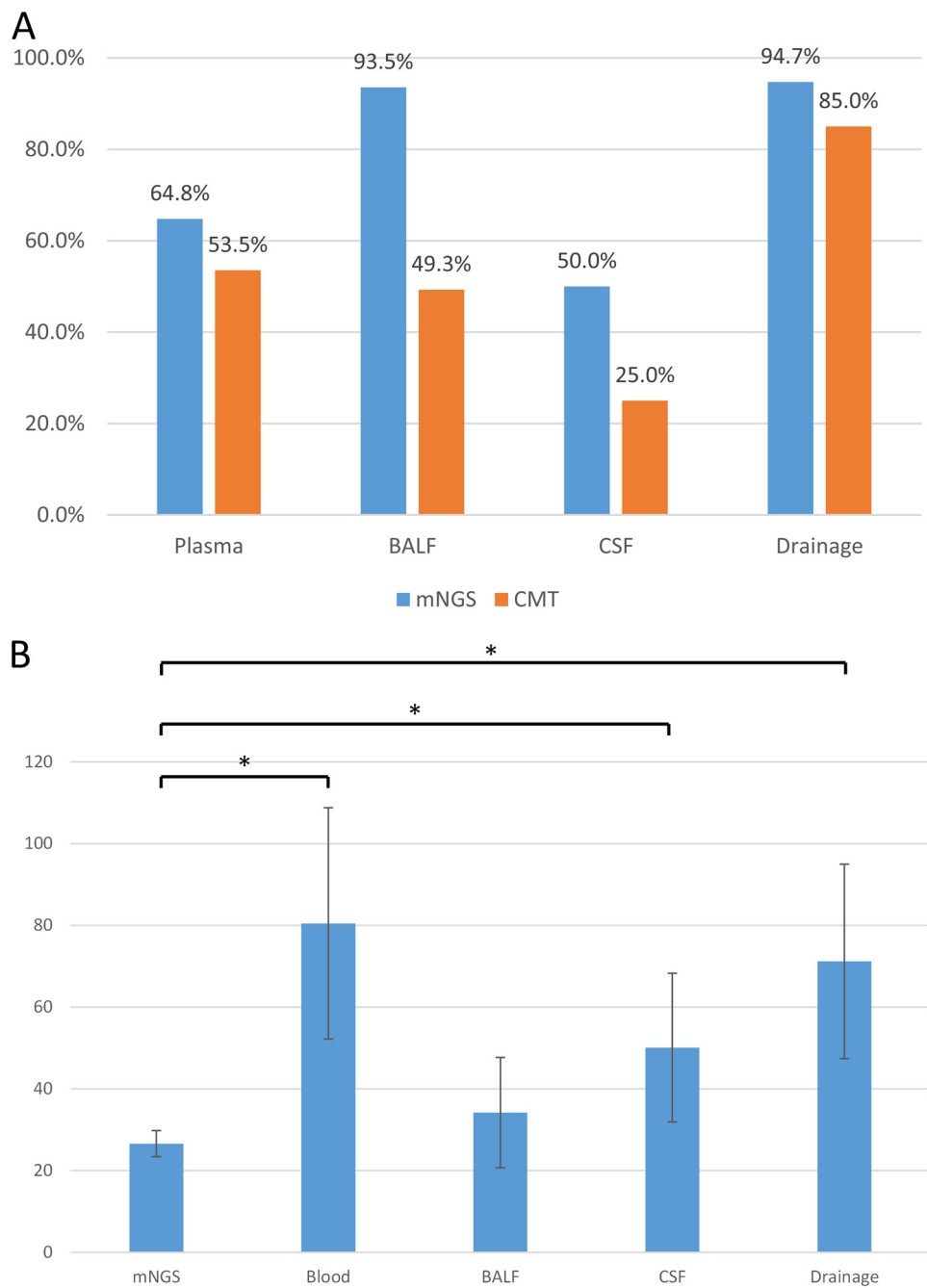

**FIG 4** Comparison of positivity rates and reporting times for different infection sites. (A) Comparison of positivity rates of mNGS and CMT for different sampling sites. (B) Comparison of reporting times (hours) between mNGS and CMT analysis for different sampling sites. mNGS, metagenomic next-generation sequencing. CMT, conventional microbiological test. BALF, bronchoalveolar lavage fluid. CSF, cerebrospinal fluid. *, $P < 0.05$.

$P = 0.005$). Lymphocyte subgroup analysis showed that T lymphocyte (577 $\pm$ 317 versus 395 $\pm$ 207, $P = 0.005$) and CD4$^+$ T lymphocyte (333 $\pm$ 199 versus 230 $\pm$ 120, $P = 0.009$) counts were lower in the matched group than the unmatched group, while the B lymphocyte, CD8$^+$ T lymphocyte, and NK cell counts did not differ ($P > 0.05$), as shown in Table S1. Inflammatory mediators were also not significantly different between the two groups ($P > 0.05$). A further comparison of the no-match, partial-match, and complete-match groups supported the above significant differences, as shown in Fig. 6. We also performed ROC curve analysis. Compared with the total lymphocyte count, the CD4$^+$ T lymphocyte count had better discriminatory ability, with an area under the concentration-time curve

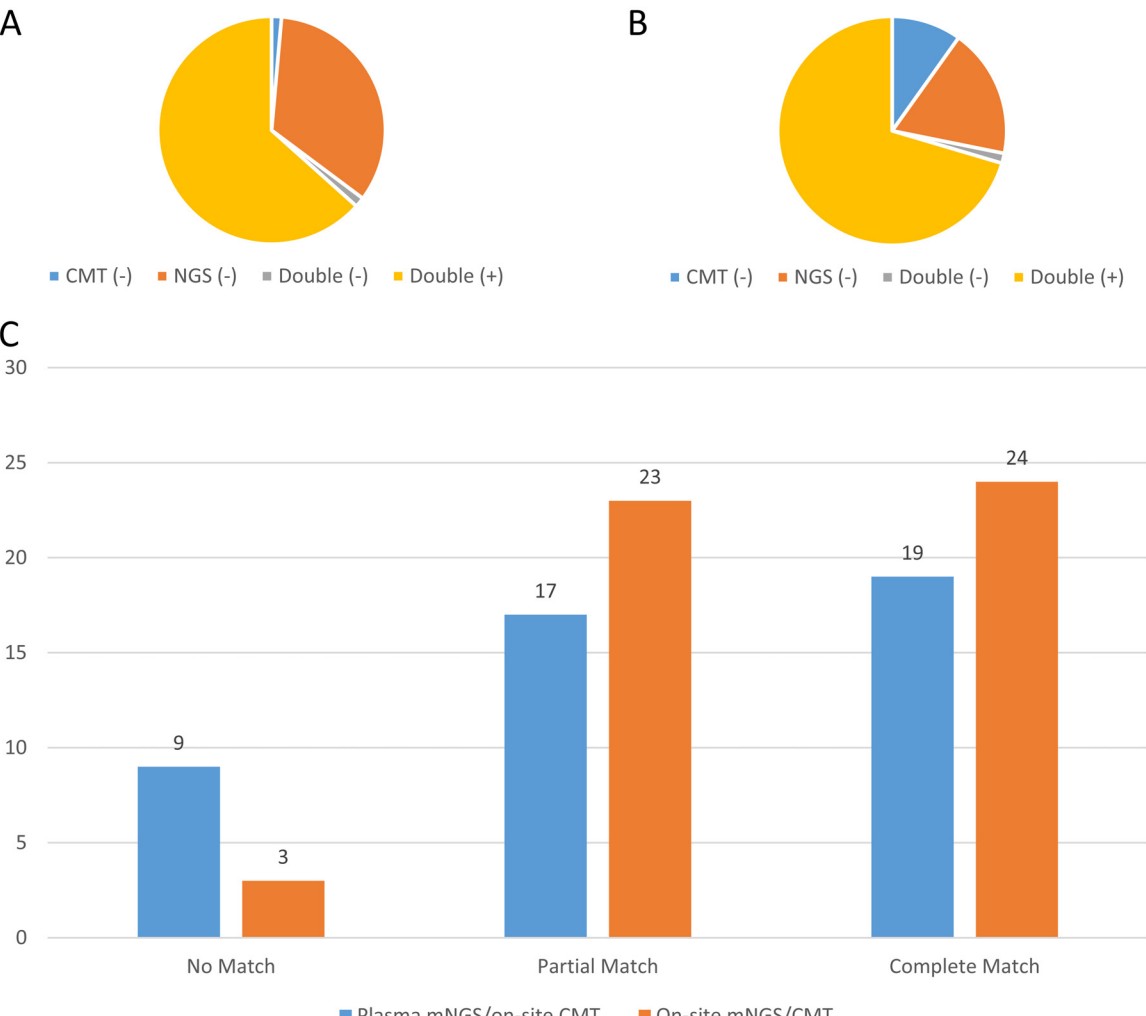

**FIG 5** Analysis of number of samples with matching plasma mNGS and CMT results. (A) Proportions of plasma mNGS and CMT results for clinically suspected infection sites. (B) Proportions of mNGS results for clinically suspected infection sites (on-site mNGS) and corresponding CMT results. (C) Concordance of mNGS results for cases with both positive plasma mNGS/on-site CMT results and on-site mNGS/CMT results.

(AUC) value of 0.752 (: 0.629, 0.875), as shown in Fig. S2. According to the ROC analysis, a CD4$^+$ T lymphocyte count lower than 266 cells/mm$^3$ was predictive of a match between plasma mNGS detection and CMT detection at clinically suspected sites.

## DISCUSSION

Sepsis is a major issue in the management of critically ill patients and is associated with high mortality rates. Studies in recent years have shown that many septic patients develop persistent and profound immunosuppression (7 to 9), and previous studies also indicated that a persistently low level of circulating lymphocytes following a diagnosis of sepsis independently predicts mortality and may serve as a biomarker for sepsis-induced immunosuppression (10, 11).

A precise and efficient microbiological diagnosis and immediate treatment are essential for sepsis patients, especially for high-risk patients with sepsis-induced lymphopenia. In recent years, a new nucleic-acid-based method, mNGS, has been widely used and has shown superior feasibility and sensitivity for pathogen detection in clinical practice. However, previous applications of mNGS have mainly focused on single categories of infection, such as encephalitis, bloodstream infections, and pneumonia (12 to 15), and a comprehensive mNGS method for the evaluation of sepsis, which may involve various infection categories, is still lacking. Considering the higher costs involved and its shortcomings when differentiating colonizing and pathogenic strains, the clinical application of mNGS, especially its interpretation, remains

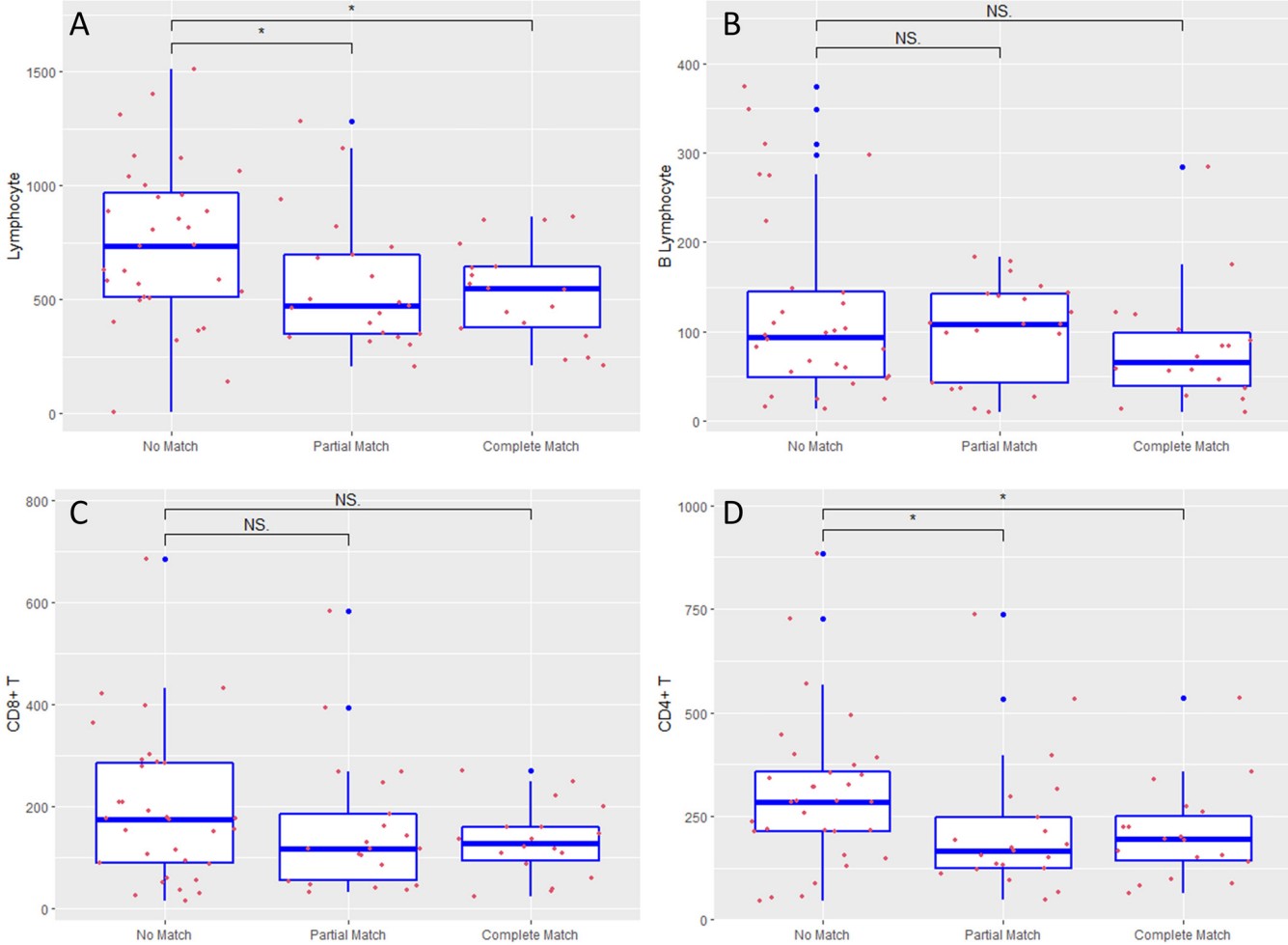

**FIG 6** Comparison of peripheral lymphocyte subset counts grouped by the accordance between plasma mNGS and CMT results for clinically suspected infection sites. The counts of total lymphocytes (A), B lymphocytes (B), CD8+ T lymphocytes (C), and CD4+ T lymphocytes (D) in patient groups withs no match, partial match, and complete match. NS, nonsignificant.

controversial (16 to 19). The controversy has increased since the sampling sources of mNGS have gradually expanded beyond plasma (16, 20).

In this study, we enrolled a cohort of 71 sepsis patients with sepsis-induced lymphopenia, for which 125 mNGS and 166 CMT tests were performed on samples from multiple clinically suspected infection sites. We analyzed the distribution of mNGS and CMT results and the matched results according to the detected strains or individual cases. The results showed that, besides the good sensitivity of mNGS detection, multiple sampling sites for both plasma and clinically suspected infection sites further improved the overall positivity rate, and further analysis showed that a CMT result matching with on-site mNGS was superior to that with plasma mNGS. To further discover the potential influencing factors associated with matching between plasma mNGS and CMT, we performed lymphocyte subset count analysis. In the groups for which plasma mNGS detection matched or partially matched the CMT results, lymphocyte counts, especially those of CD4+ T lymphocytes, were significantly lower than those in the mismatch group.

Previous studies showed that a prominent decline in lymphocyte (specifically CD4+ T lymphocytes) counts may be observed in sepsis (21, 22), and sepsis-associated lymphopenia is significantly associated with patients who die from sepsis (10). Thus, compared with immunocompetent patients, patients with sepsis-associated lymphopenia may present some distinct features during ICU admission. First, as a result of their immunocompromised state, polymicrobial and multisite infections frequently occur (16), but with the gradual maturity and

promotion of mNGS detection, multisite sampling has become a possibility. Second, in immunocompromised septic patients, rare, atypical, or nonpathogenic microorganisms are frequently detected, and it is a major challenge for intensivists to distinguish the pathogen of interest from others and implement a targeted treatment plan (23). Third, the treatment of sepsis requires rapid diagnosis and correct treatment, especially for patients with septic-associated lymphopenia, who have been demonstrated to be associated with a poor prognosis. Thus, rapidly screening patients who may benefit from mNGS detection would be beneficial in clinical practice.

To improve the efficiency and accuracy of diagnoses based on mNGS in this study, we assessed the significance of multisite mNGS detection on pathogen identification. To our knowledge, this was the first study to describe the application of multisite mNGS for sepsis patients and to evaluate its diagnostic performance in combination with lymphocyte subgroup analysis, which has become a fundamental method for immune monitoring of sepsis cases (9). We demonstrated that, based on plasma mNGS, additional mNGS sampling of the clinically suspected sites may improve the clinical significance of the results; on the other hand, for patients with distinct lymphopenia, especially decreased $CD4^+$ T lymphocytes, plasma mNGS may provide a more reliable identification of pathogenic microorganisms. Our results further promote the application of plasma mNGS, which was conventionally believed to only be useful for bloodstream infections (23 to 25). Considering the relatively high cost of multisite mNGS detection, lymphocytes subset counts can still contribute to the interpretation of plasma mNGS detection results. This finding is consistent with our previous results on the clinical significance of lymphocyte subset analysis (26, 27) and highlights the importance of the early recognition, diagnosis, and treatment of sepsis-associated lymphopenia.

Despite these promising results, our study had some limitations. First, we enrolled a relatively small observational cohort. Second, for patients with sepsis-associated lymphopenia, it may not be easy to definitively ascertain the pathogen in every case. The matching between plasma mNGS and multisite CMT detection at clinically suspected sites was evaluated to represent whether the "real" causative pathogen was detected, but this may have underestimated the sensitivity of mNGS. Finally, considering that mNGS detection may be influenced by various factors, the results obtained in our single-center study should be validated before applying to other centers.

**Conclusion.** In this prospective study of intubated sepsis-associated lymphopenia patients in an ICU, we found that mNGS tests of plasma and other clinically suspected sites showed a higher positivity rate than CMT detection. The CMT results for multiple clinically suspected sites were more closely matched to the multisite mNGS results than to the plasma mNGS results, whereas, with only plasma mNGS, a lower $CD4^+$ T lymphocyte count ($\leq$266 cells/mm³) predicted a better match between plasma mNGS and multisite CMT detection. Our findings highlighted the importance of $CD4^+$ T lymphocyte counts and potential sepsis-associated lymphopenia in the diagnosis and treatment of sepsis and support the addition of lymphocyte subset analysis to the clinical interpretation of mNGS results.

## MATERIALS AND METHODS

**Study design and participants.** Intubated patients who were admitted to the ICU of the Peking Union Medical College Hospital (PUMCH) from January 2020 to February 2022 were recruited. The inclusion criteria were (i) age $\geq$18 years, (ii) ICU stay of >24 h, (iii) diagnosis with sepsis 3.0 after reference to international diagnostic criteria (28), and (iv) new-onset lymphopenia ($\leq$1.2 $\times$ 10⁹/L) after sepsis diagnosis (10). The exclusion criteria were (i) incomplete clinical data, including microbiological data; (ii) any condition causing primary or acquired immunocompromised state, including diagnosis of hematological or immunological disease, and treatment with chemotherapy agents or corticosteroids within 6 months prior to ICU admission; (ii) failure to provide a sample for mNGS analysis during the first 24 h after ICU admission; (iv) a life expectancy of <24 h; and (v) failure to provide written consent. The study was designed and carried out in accordance with the principles of the Declaration of Helsinki. This study was approved by the institutional review board of Peking Union Medical College Hospital (approval number: JS-1170; study title: "Clinical and Fundamental Investigation on the Impact of mTOR Signaling Pathway Mediating $CD8^+$ Tem Cell Proliferation on Improving Immunity Response to Sepsis"; approval date: November 2016). Informed consent was obtained from all patients involved, and the study was registered at http://www.chictr.org.cn/ (identifier ChiCTR-ROC-17010750).

**Data collection.** In this study, patient demographics, clinical data (including infection category and source), Acute Physiology and Chronic Health Evaluation (APACHE) II score, Sequential Organ Failure Assessment (SOFA) score, and in-hospital mortality were recorded. During the first 24 h after ICU admission, peripheral blood samples were obtained for mNGS analysis, and samples from other clinically suspected infection sites were also obtained for both mNGS and conventional rapid examination (Gram-staining smear) and culture. For patients with clinically suspected pneumonia, BALF was collected for both mNGS and culture, while for other patients, sputum was collected through trachea cannula for culture to exclude potential pulmonary infection. Peripheral blood culture was collected for all enrolled patients to diagnose bloodstream infections, and the time to initial positive detection and report was recorded. CMT, including bacterial/fungal stains and cultures on peripheral blood, BALF and other sterile body fluids, single or multiplex RT-PCR, blood culture, serum, and urine pathogen-specific antigen tests, and serum pathogen-specific antibody tests, was prescribed and conducted according to a clinical assessment of necessity and clinical guidelines for the corresponding infectious diseases. The time from sampling to reporting meaningful result was also recorded, which means time to report culture result (positivity or negativity) for BALF, CSF and drainage fluid. Culture and confirmation of species identification were performed at the central laboratory of the Clinical Laboratory Department, PUMCH, by matrix-assisted laser desorption/ionization time-of-flight mass spectrometry (Vitek MS; bioMérieux, Marcy l'Etoile, France).

**Lymphocyte subpopulation and serum inflammation markers.** At ICU admission, peripheral blood samples were also obtained for a routine examination that included complete blood counts, C-reactive protein, procalcitonin, and immunological parameters, which were measured in the PUMCH laboratories, as previously described (29, 30). In brief, freshly collected EDTA anticoagulated whole blood was incubated and tested with a panel of monoclonal antibodies, then subjected to flow cytometric analysis using a three-color EPICS-XL flow cytometer (Beckman Coulter, Brea, CA, USA) to detect T cells (CD3$^+$), CD4$^+$ and CD8$^+$ T-cell subgroups, B-cells (CD19$^+$), and natural killer (NK) cells (CD3$-$CD16$^+$ CD56$^+$). Rate nephelometry (Array 360; Beckman Coulter, Brea, CA, USA) was used to measure serum levels of immunoglobulin (Ig)A, IgG, and IgM and of complement factors C3 and C4.

**Metagenomic next-generation sequencing and data analysis.** After each sample was obtained, mNGS was performed via the following steps, as described previously (31).

**Sample collection and DNA extraction.** After centrifugation at 1,900 $\times$ $g$ for 10 min at 4°C, DNA was extracted with the PathoXtract Plasma Nucleic Acid kit (WYXM03001S; Willingmed Corp., Beijing, China) from the supernatant of collected samples.

**Library construction, sequencing, and data analysis.** Libraries for NGS were prepared from cell-free DNA using the KAPA DNA HyperPrep kit (KK8504; Kapa Biosystems, Wilmington, MA, USA) in accordance with the manufacturer's protocol. Sequencing of the libraries was performed on NextSeq 550Dx (Illumina), and at least 25 million sequencing reads were acquired for each sample.

**Pipeline of bioinformatics analysis.** Genomic data for bacteria, fungi, viruses, parasites, archaea, and other pathogenic microorganisms were obtained from NCBI (https://ftp.ncbi.nlm.nih.gov/genomes/). Sequencing data were processed automatically using the Pathogen Identification Sequencing Metagenomic Sequencing Data Management System V2.0 (Willingmed Corp.). After filtration of low-quality sequences, contaminated sequences, high-coverage repeats, or short read-length sequences, the remaining high-quality sequencing data were compared with the human reference genome GRCH37 (hg19) to remove human host sequences. Then, the cleaned data were used in subsequent identification and analyses. For the identification of pathogens, an RPTM value, defined as the number of pathogen-specific reads per 10 million (RPTM), was used to positively identify pathogens. A RPTM threshold of $\geq$5 for bacteria and fungi was set for the positive detection of a pathogen.

**Statistical analysis.** Statistical analysis was conducted using IBM SPSS v24.0 software (IBM SPSS, Armonk, NY, USA). Measurement data were expressed as the mean $\pm$ standard deviation (SD), median, and interquartile range, or proportions, as appropriate. The Student's $t$ test or Mann-Whitney U test was used to compare continuous variables; the $\chi^2$ test or Fisher's exact test was used to compare categorical variables. Differences with values of $P < 0.05$ were considered statistically significant.

**Data availability.** The data sets presented in this study can be found in online repositories at https://www.ncbi.nlm.nih.gov/, via BioProject accession number PRJNA850156.

## SUPPLEMENTAL MATERIAL

Supplemental material is available online only.

**SUPPLEMENTAL FILE 1**, PDF file, 0.4 MB.

## ACKNOWLEDGMENTS

The work was supported by the National Natural Science Foundation of China (no. 82072226), National Key R&D Program of China 2022YFC2009803 from the Ministry of Science and Technology of the People's Republic of China, Beijing Municipal Science and Technology Commission (no. Z201100005520049), and CAMS Innovation Fund for Medical Sciences (CIFMS) 2021-I2M-1-062 from the Chinese Academy of Medical Sciences.

The authors report no conflicts of interest. All authors have submitted the ICMJE Form for Disclosure of Potential Conflicts of Interest. Conflicts that the editors consider relevant to the content of the manuscript have been disclosed.

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
