## [Reviewer comments · Microbiology Spectrum]

Microbiology Spectrum

Multisite metagenomic next-generation sequencing improved diagnostic performance for sepsis-associated lymphopenia patients

Dongkai Li, Wei Gai, Jiahui Zhang, Wei Cheng, Na Cui, and Hao Wang

Corresponding Author(s): Na Cui, Peking Union Medical College Hospital

Review Timeline:

Submission Date:	September 1, 2022
Editorial Decision:	October 11, 2022
Revision Received:	October 24, 2022
Editorial Decision:	November 1, 2022
Revision Received:	November 2, 2022
Accepted:	November 6, 2022

Editor: Rosemary She

Reviewer(s): The reviewers have opted to remain anonymous.

Transaction Report:

DOI: <https://doi.org/10.1128/spectrum.03532-22>

October 11, 2022

Dr. Na Cui
Peking Union Medical College Hospital
Critical Care Medicine
1 Shuaifuyuan, Dongcheng District, Beijing
Beijing
China

Re: Spectrum03532-22 (Multisite metagenomic next-generation sequencing improved diagnostic performance for sepsis-associated lymphopenia patients)

Dear Dr. Na Cui:

Thank you for submitting your manuscript to Microbiology Spectrum. Your manuscript has been peer-reviewed and the consensus decision is Modifications. There were major concerns regarding the limited information provided for the comparator, conventional microbiology tests. When submitting the revised version of your paper, please provide (1) point-by-point responses to the issues raised by the reviewers as file type "Response to Reviewers," not in your cover letter, and (2) a PDF file that indicates the changes from the original submission (by highlighting or underlining the changes) as file type "Marked Up Manuscript - For Review Only". Please use this link to submit your revised manuscript - we strongly recommend that you submit your paper within the next 60 days or reach out to me. Detailed instructions on submitting your revised paper are below.

Link Not Available

Sincerely,

Rosemary She

Journals Department
Reviewer comments:

Reviewer #1 (Comments for the Author):

Major Comments for the Author

In this manuscript, Li et al. endeavored to evaluate the role of multisite mNGS in diagnostic performance for sepsis associated lymphopenia patients. There are numerous questions that are needed to be clarified by the authors. The reviewer raised the following concerns and questions:

1. Why did the exclusion criteria include primary or acquired immunocompromised state?

2. In their mNGS, cell free DNA are detected so this method cannot detect RNA viruses. The authors should point it out to avoid confusion to readers. Also the fact that this assay is only targeting cell free DNA also may explain discrepancy between mNGS and traditional micro tests.
3. What is the definition and content of CMTs? Do CMT include blood culture and BCID multiplex PCR panel? Without clear definition of CMT, the gold standard, it is hard to evaluate the manuscript.
4. Some important index calculation are missing. The authors need to analyze the PPA, NPA, OPA between the mNGS and CMT

Minor Comments for Author (Required)

None

Reviewer #2 (Comments for the Author):

Major Comments:

Materials and methods: Overall I think there needs to be some clarity as to what specimens were collected and what was performed with each one. It seems like blood culture was done on all subjects and then BALs from some along with a random set of other possible sites. Was all mNGS from direct specimens or from positive blood culture bottles. For the conventional culture what and how was it performed. It is impossible to compare the results if its unclear the conventional testing methods. Why was there a large difference of specimens tested between culture and mNGS? Was this based on physician orders? Especially as there were more BAL collected for culture, it is surprising that not all of those were sent for mNGS (if that was a research specific test).

Figure 4B. Were reporting times performed for all specimens or just positives. Looking at the times for CMT, some of those seem like they would be specific to including negatives as well, which would hold cultures for later. This data might be better as time to positive results. Also, without knowing the full CMT its difficult to interpret these results. For instance, are techs performing MALDI-ID or having to run a biochemical panel, which would add on another 12+ hours.

For specimens that were mNGS positive and culture negative, were any subjects on antibiotics prior to specimen collection? Culture is limited by viable organisms and this difference may be based on antimicrobial therapy and not due to increased sensitivity of mNGS.

Minor comments:

Ln138-139 was these only defined based on CMT or is this also from mNGS?

Figure 3: As you collected specimens from multiple sites, how often were the same pathogens detected from the various sources?

Ln166 misspelled suspected

Staff Comments:

Preparing Revision Guidelines

Please return the manuscript within 60 days; if you cannot complete the modification within this time period, please contact me. If you do not wish to modify the manuscript and prefer to submit it to another journal, please notify me of your decision immediately so that the manuscript may be formally withdrawn from consideration by Microbiology Spectrum.

The manuscript submitted by Li et al. and entitled “Multisite Metagenomic next-generation sequencing improved diagnostic performance for sepsis-associated lymphopenia patients” describes a study using multisite mNGS for the detection of sepsis. In this work, an observational study using intubated lymphopenia patients was performed for 2 years. In this period 71 patients were enrolled and collected, and testing was performed from blood, BALs, CSF, and drainage. mNGS detected 95 positive pathogens compared to 91 from conventional culture. The study is interesting and demonstrates some needed data for the use of mNGS and possibility of testing multiple sites from these patients. However, there are several topics that need to be addressed to fully evaluate the work and except for publication. Below are my suggested comments and questions.

Major Comments:

Materials and methods: Overall I think there needs to be some clarity as to what specimens were collected and what was performed with each one. It seems like blood culture was done on all subjects and then BALs from some along with a random set of other possible sites. Was all mNGS from direct specimens or from positive blood culture bottles. For the conventional culture what and how was it performed. It is impossible to compare the results if its unclear the conventional testing methods.

Why was there a large difference of specimens tested between culture and mNGS? Was this based on physician orders? Especially as there were more BAL collected for culture, it is surprising that not all of those were sent for mNGS (if that was a research specific test).

Figure 4B. Were reporting times performed for all specimens or just positives. Looking at the times for CMT, some of those seem like they would be specific to including negatives as well, which would hold cultures for later. This data might be better as time to positive results. Also, without knowing the full CMT its difficult to interpret these results. For instance, are techs performing MALDI-ID or having to run a biochemical panel, which would add on another 12+ hours.

For specimens that were mNGS positive and culture negative, were any subjects on antibiotics prior to specimen collection? Culture is limited by viable organisms and this difference may be based on antimicrobial therapy and not due to increased sensitivity of mNGS.

Minor comments:

Ln138-139 was these only defined based on CMT or is this also from mNGS?

Figure 3: As you collected specimens from multiple sites, how often were the same pathogens detected from the various sources?

Ln166 misspelled suspected

Reviewer comments:

Reviewer #1 (Comments for the Author):

Major Comments for the Author

In this manuscript, Li et al. endeavored to evaluate the role of multisite mNGS in diagnostic performance for sepsis associated lymphopenia patients. There are numerous questions that are needed to be clarified by the authors. The reviewer raised the following concerns and questions:

1. Why did the exclusion criteria include primary or acquired immunocompromised state?

Thank you for your comments. Considering the background of this study involved the sepsis-associated lymphopenia, the primary or acquired immunocompromised patients should be excluded to match its diagnosis. In fact, the primary or acquired immunocompromised state may influence the balance of lymphocytes and other immune cells and lead to incorrect conclusion on the association between the immune state and the mNGS detection.

2. In their mNGS, cell free DNA are detected so this method cannot detected RNA viruses. The authors should point it out to avoid confusion to readers. Also the fact that this assay is only targeting cell free DNA also may explain discrepancy between mNGS and traditional micro tests.

Thank you for your comments. Essentially, mNGS is a technique of cell-free fluid based nucleic acid detection while some progress had been made in the whole cell detection [1]. Only DNA detection was performed and recorded in the study because virus, especially for RNA virus associated sepsis was rare in the critical care settings of China, and importantly, the nucleic acid detection on virus cannot be compared with the CMT method, which is mainly a culture method of bacteria and fungi. We had made some necessary clarification in the revised manuscript.

[1] Yu L, Zhang Y, Zhou J, Zhang Y, Qi X, Bai K, Lou Z, Li Y, Xia H, Bu H. Metagenomic next-generation sequencing of cell-free and whole-cell DNA in diagnosing central nervous system infections. *Front Cell Infect Microbiol.* 2022 Sep 27;12:951703. doi: 10.3389/fcimb.2022.951703. PMID: 36237422; PMCID: PMC9551220.

3. What is the definition and content of CMTs? Do CMT include blood culture and BCID multiplex PCR panel? Without clear definition of CMT, the gold standard, it is hard to evaluate the manuscript.

Thank you for your comments. Basically, the CMT included blood/sputum/BALF/CSF cultures, serological tests, molecular diagnostic tests, and antigen detection [1-2]. The clear definition

was added in the revised manuscript.

1. Fan S, Si M, Xu N, Yan M, Pang M, Liu G, Gong J, Wang H. Metagenomic next-generation sequencing-guided antimicrobial treatment versus conventional antimicrobial treatment in early severe community-acquired pneumonia among immunocompromised patients (MATESHIP): A study protocol. *Front Microbiol.* 2022 Aug 2;13:927842. doi: 10.3389/fmicb.2022.927842. PMID: 35983331; PMCID: PMC9379097.
2. Zheng YR, Lin SH, Chen YK, Cao H, Chen Q. Application of metagenomic next-generation sequencing in the detection of pathogens in bronchoalveolar lavage fluid of infants with severe pneumonia after congenital heart surgery. *Front Microbiol.* 2022 Aug 5;13:954538. doi: 10.3389/fmicb.2022.954538. PMID: 35992666; PMCID: PMC9391048.

4. Some important index calculation are missing. The authors need to analyze the PPA, NPA, OPA between the mNGS and CMT

Thank you for your comments. As stated above, not all enrolled patients underwent the mNGS and CMT on the same site concurrently except for the plasma mNGS and peripheral blood culture. Considering the imbalance among groups and small number of cases, we believed its significance was limited. For reference, we listed the positivity data and agreement results on blood and pulmonary sites of the two methods as below.

Blood culture and Plasma mNGS			
	CMT+	CMT-	
NGS+	31	15	46
NGS-	7	18	25
Total	38	43	71

BALF culture and mNGS			
	CMT+	CMT-	
NGS+	20	9	29
NGS-	0	2	2
Total	20	11	31

Index	Blood culture and Plasma mNGS	BALF culture and mNGS
--------------	--------------------------------------	------------------------------

positive percent agreement, PPA	77.5%	100.0%
negative percent agreement, NPA	54.5%	18.2%
overall percent agreement, OPA	69.0%	71.0%

Minor Comments for Author (Required)

None

Reviewer #2 (Comments for the Author):

Major Comments:

Materials and methods: Overall I think there needs to be some clarity as to what specimens were collected and what was performed with each one. It seems like blood culture was done on all subjects and then BALs from some along with a random set of other possible sites. Was all mNGS from direct specimens or from positive blood culture bottles. For the conventional culture what and how was it performed. It is impossible to compare the results if its unclear the conventional testing methods.

Thank you for your comments. As stated in the Method section, all patients underwent plasma mNGS at ICU admission, which was performed on the peripheral blood and irrelevant with the blood culture. For patients with clinically suspected pneumonia, bronchoalveolar lavage fluid (BALF, by bronchofiberscope procedure) was collected for mNGS and CMT, while for other patients, sputum was collected through trachea cannula for culture. The definition and procedure of conventional microbiological method was clarified in the revised manuscript.

Why was there a large difference of specimens tested between culture and mNGS? Was this based on physician orders? Especially as there were more BAL collected for culture, it is surprising that not all of those were sent for mNGS (if that was a research specific test).

Thank you for your comments. As stated above, all patients underwent plasma mNGS at ICU admission while only patients with clinically suspected pneumonia, BALF (by bronchofiberscope procedure) was collected. The mNGS detection is based on plasma, BALF (instead of sputum) or other sterile body fluid. Considering the costs and procedure related risk, we cannot require all patients underwent BALF mNGS detection. Instead, sputum culture could be performed to screen potential pulmonary infection. In fact, analyzing the value of plasma mNGS on discovering infections beyond the bloodstream was one of the main topics in our study.

Figure 4B. Were reporting times performed for all specimens or just positives. Looking

at the times for CMT, some of those seem like they would be specific to including negatives as well, which would hold cultures for later. This data might be better as time to positive results. Also, without knowing the full CMT its difficult to interpret these results. For instance, are techs performing MALDI-ID or having to run a biochemical panel, which would add on another 12+ hours.

Thank you for your comments. The Fig 4B compared the reporting time of different detection methods. From the viewpoint of clinical physicians, only the time from sampling to reporting meaningful result was calculated, which means time to positivity for blood culture and time to report culture result (positivity or negativity) for BALF, CSF and CSF.

For specimens that were mNGS positive and culture negative, were any subjects on antibiotics prior to specimen collection? Culture is limited by viable organisms and this difference may be based on antimicrobial therapy and not due to increased sensitivity of mNGS.

Thank you for your comments. Considering most of the patients at ICU admission were post-surgical or transferred from Emergency department, previous antibiotics treatment was inevitable and such situation on sepsis were similar for clinical studies in ICU. However, as stated in the Method section, we kept all patients underwent necessary CMT and mNGS detection during the first 24 hours after ICU admission to reduce the potential impact from previous treatment. Negative culture result from antibiotic treatment is an important clinical issue because it would impede the identification of pathogen and selection of antibiotics. In such circumstances, mNGS, of which the result is irrelevant with whether the pathogen is viable, is of great significance in clinical practice.

Minor comments:

Ln138-139 was these only defined based on CMT or is this also from mNGS?

Figure 3: As you collected specimens from multiple sites, how often were the same pathogens detected from the various sources?

Ln166 misspelled suspected

Thank you for your comments. The patients' diagnosis was based on clinical criteria of each infection category, which include CMT detection when necessary. Generally, the mNGS test was not part of the diagnosis criteria.

The same pathogen detected from various source may be caused by multiple site colonization, bloodstream infection migrated from other infection sites, or cross infection. In our study, such patients consist of less than 10%.

The typos were corrected in the revised manuscript.

November 1, 2022

Dr. Na Cui
Peking Union Medical College Hospital
Critical Care Medicine
1 Shuaifuyuan, Dongcheng District, Beijing
Beijing
China

Re: Spectrum03532-22R1 (Multisite metagenomic next-generation sequencing improved diagnostic performance for sepsis-associated lymphopenia patients)

Dear Dr. Na Cui:

Your response to reviewer comments have been evaluated and while the remarks have been partially addressed, several issues require further attention. I am hopeful that you will be able to provide an adequate response to the following reviewer comments as listed below.

1. Thank you for addressing the reviewer comment to add definition and content of CMTs. Please correct the sentence you have added on page 7 of the marked-up manuscript. "Multiple PCR" - is this supposed to be "Multiplex PCR"? Also, "...underwent according to the consensus of the corresponding infection category" needs to be clarified and checked for English grammar.
2. Regarding the Reviewer comment, "Some important index calculation are missing. The authors need to analyze the PPA, NPA, OPA between the mNGS and CMT," I thank you for providing the tables in the rebuttal. I agree with the reviewer that these are significant calculations that should be included in the manuscript. Please add 95% confidence intervals and include the Tables as Supplemental Data.
3. Reviewer 2 requests more information "for the conventional culture what and how was it performed." Please include at minimum the blood culture system that was used and the standard identification method of the Clinical Laboratory (biochemical vs. MALDI-TOF vs. other) as this is essential to understanding the turnaround time information in the Results and Figure 4B.
4. The response to "Were reporting times performed for all specimens or just positives" was provided in the rebuttal ("only the time from sampling to reporting meaningful result was calculated, which means time to positivity for blood culture and time to report culture result (positivity or negativity) for BALF, CSF and CSF"). Please also include this information in the manuscript, at a minimum to the Results on page 10 of marked-up manuscript.

Link Not Available

Sincerely,

Rosemary She

Journals Department
Reviewer comments:

Staff Comments:

Preparing Revision Guidelines

Please return the manuscript within 60 days; if you cannot complete the modification within this time period, please contact me. If you do not wish to modify the manuscript and prefer to submit it to another journal, please notify me of your decision immediately so that the manuscript may be formally withdrawn from consideration by Microbiology Spectrum.

- 1. Thank you for addressing the reviewer comment to add definition and content of CMTs. Please correct the sentence you have added on page 7 of the marked-up manuscript. "Multiple PCR" - is this supposed to be "Multiplex PCR"? Also, "...underwent according to the consensus of the corresponding infection category" needs to be clarified and checked for English grammar.**

Thank you. The "'Multiple PCR" and related statement were corrected in the revised manuscript.

- 2. Regarding the Reviewer comment, "Some important index calculation are missing. The authors need to analyze the PPA, NPA, OPA between the mNGS and CMT," I thank you for providing the tables in the rebuttal. I agree with the reviewer that these are significant calculations that should be included in the manuscript. Please add 95% confidence intervals and include the Tables as Supplemental Data.**

Thank you. The comparison between mNGS and CMT as well as the agreement calculation were included in the revised Supplement Table 2 and Table 3.

- 3. Reviewer 2 requests more information "for the conventional culture what and how was it performed." Please include at minimum the blood culture system that was used and the standard identification method of the Clinical Laboratory (biochemical vs. MALDI-TOF vs. other) as this is essential to understanding the turnaround time information in the Results and Figure 4B.**

Thank you. The equipment for culture and identification was stated in the Method part of the revised manuscript.

- 4. The response to "Were reporting times performed for all specimens or just positives" was provided in the rebuttal ("only the time from sampling to reporting meaningful result was calculated, which means time to positivity for blood culture and time to report culture result (positivity or negativity) for BALF, CSF and CSF"). Please also include this information in the manuscript, at a minimum to the Results on page 10 of marked-up manuscript.**

Thank you. The corresponding information, of which the blood culture had been described, was added to the Method part of the revised manuscript.

November 6, 2022

Dr. Na Cui
Peking Union Medical College Hospital
Critical Care Medicine
1 Shuaifuyuan, Dongcheng District, Beijing
Beijing
China

Re: Spectrum03532-22R2 (Multisite metagenomic next-generation sequencing improved diagnostic performance for sepsis-associated lymphopenia patients)

Dear Dr. Na Cui:

Your manuscript has been accepted, and I am forwarding it to the ASM Journals Department for publication. You will be notified when your proofs are ready to be viewed.

Sincerely,

Rosemary She
Editor, Microbiology Spectrum
